# Generalization bounds for mixing processes
# via delayed online-to-PAC conversions

**Baptiste Abélès**                                    BAPTISTE.ABELES@GMAIL.COM
*Universitat Pompeu Fabra, Barcelona*

**Eugenio Clerico**                                    EUGENIO.CLERICO@GMAIL.COM
*Universitat Pompeu Fabra, Barcelona*

**Gergely Neu**                                        GERGELY.NEU@GMAIL.COM
*Universitat Pompeu Fabra, Barcelona*

**Editors:** Gautam Kamath and Po-Ling Loh

## Abstract

We study the generalization error of statistical learning algorithms in a non i.i.d. setting, where the training data is sampled from a stationary mixing process. We develop an analytic framework for this scenario based on a reduction to online learning with delayed feedback. In particular, we show that the existence of an online learning algorithm with bounded regret (against a fixed statistical learning algorithm in a specially constructed game of online learning with delayed feedback) implies low generalization error of said statistical learning method even if the data sequence is sampled from a mixing time series. The rates demonstrate a trade-off between the amount of delay in the online learning game and the degree of dependence between consecutive data points, with near-optimal rates recovered in a number of well-studied settings when the delay is tuned appropriately as a function of the mixing time of the process.

## 1. Introduction

In machine learning, generalization means the ability of a model to infer patterns from a dataset of training examples and apply them to analyze previously unseen data (Shalev-Shwartz and Ben-David, 2014). The gap in accuracy between the model's predictions on new data and those on the training set is usually referred to as *generalization error*. Providing upper bounds on this quantity is a central goal in statistical learning theory. Classically, bounds based on notions of complexity of the model's hypothesis space (e.g., its VC dimension or Rademacher complexity) were used to provide uniform worst-case guarantees (see Bousquet et al., 2004; Vapnik, 2013; Shalev-Shwartz and Ben-David, 2014). However, these results are often too loose to be applied to over-parameterised models such as deep neural networks (Zhang et al., 2021). As a consequence, several approaches have been proposed to obtain algorithm-dependent generalization bounds, which can adapt to the problem and be much tighter in practice than their uniform counterparts. Often, the underlying idea is that if the algorithm's output does not have a too strong dependence on the specific input dataset used for the training, then the model should not be prone to overfitting, and thus generalize well. Examples of results that build onto these ideas are stability bounds, information-theoretic bounds, and PAC-Bayesian bounds (see, e.g., Bousquet and Elisseeff, 2002; Russo and Zou, 2020; Hellström et al., 2023; Alquier, 2024).

Most results in the literature focus on the i.i.d. setting, where the training dataset is made of independent draws from some underlying data distribution. However, for several applications, this assumption is far from realistic. For instance, it excludes the case where observations received by

the learner have some inherent temporal dependence, as it is the case for stock prices, daily energy consumption, or sensor data from physical environments (Ariyo et al., 2014; Takeda et al., 2016). This calls for the development of theory for addressing non-i.i.d. data. A common approach in the extant literature is to consider a class of non-i.i.d. data-generating processes usually referred to as stationary $\beta$-mixing or $\varphi$-mixing processes. This assumption, together with a "blocking" trick introduced by Yu (1994), has led to a few results in the literature: Meir (2000), Mohri and Rostamizadeh (2008), Shalizi and Kontorovich (2013), and Wolfer and Kontorovich (2019) provided uniform worst-case generalization bounds, Steinwart and Christmann (2009) and Agarwal and Duchi (2012) discussed excess risk bounds (comparing the algorithm's output with the best possible hypothesis), while Mohri and Rostamizadeh (2010) gave bounds based on a stability analysis (in the sense of Bousquet and Elisseeff, 2002).

In the present paper, we propose a new framework for proving generalization bounds for the non-i.i.d. setting, which take a form that is similar in spirit to PAC-Bayesian bounds (Guedj, 2019; Alquier, 2024): high-probability upper bounds on the expected generalization error of randomized learning algorithms. We achieve this by combining the blocking argument by Yu (1994) to manage the concentration of sums of correlated random variables, with the recent *online-to-PAC conversion* technique recently proposed by Lugosi and Neu (2023). Using their framework we show a new way to obtain generalization bounds for stationary dependent processes that satisfy a certain "short-memory" property, intuitively meaning that data points that are closer in time are more heavily dependent on each other. Our assumption slightly relaxes the commonly considered $\beta$-mixing condition in that we only need it to hold for a specific class of bounded loss functions (as opposed to requiring stronger conditions phrased in terms of total variation distance). Among other results, this allows us to prove PAC-Bayesian generalization bounds for mixing processes. This complements previous work on such bounds that have largely considered mild relaxations of the i.i.d. condition such as assuming that the data has a martingale structure (e.g., Seldin et al., 2012; Chugg et al., 2023; Haddouche and Guedj, 2023). A notable exception is the work of Ralaivola et al. (2010), who considered data sets whose dependence structure is described via graphs, and proved generalization bounds that scale with various graph properties describing the overall strength of dependencies. One of the applications of their results is a generalization bound for $\beta$-mixing processes, comparable in nature to some of our results. We return to discussing the similarities and differences in Section 6. Further relevant works are those of Alquier and Wintenberger (2012), Alquier et al. (2013), and Eringis et al. (2022, 2024), who provided generalization bounds for a sequential prediction setting where both the data-generating process and the hypothesis class used for prediction are stable dynamical systems. Their results are proved under some very specific conditions on these systems, and their guarantees involve unspecified problem-dependent constants that may be large. In contrast, our bounds hold under general, simple-to-verify conditions and feature explicit constants.

The rest of the paper is organized as follows. In Section 2 we properly define the generalization error of a statistical learning algorithm for both i.i.d. and non-i.i.d. cases, and state our main assumption on the data dependence. Our main contribution lies in Section 3, where after recalling the results of Lugosi and Neu (2023) for the i.i.d. setting we show how to adapt their technique to stationary mixing processes. In Section 4 we provide some concrete results of the bounds we can obtain through the resulting online-to-PAC conversion methodology. Finally in Section 5 we extend our results to the setting where the hypothesis class itself may consist of dynamical systems.

**Notation.** For a distribution over hypotheses $P \in \Delta_{\mathcal{W}}$ and bounded function $f : \mathcal{W} \to \mathbb{R}$ we write $\langle P, f \rangle$ to refer to the expectation of $\mathbb{E}_{W \sim P}[f(W)]$. We denote $\mathcal{D}_{\mathrm{KL}}(P \| Q) = \mathbb{E}_{X \sim P} \left[ \ln \left( \frac{\mathrm{d}P}{\mathrm{d}Q}(X) \right) \right]$

to refer to the Kullback–Leibler divergence. We use $\|\cdot\|$ to denote a norm on the Banach space $\mathcal{Q}$ of the finite signed measures, and $\|\cdot\|_*$ the corresponding dual norm on the dual space $\mathcal{Q}^*$ of measurable functions $f$ on $\mathcal{W}$ such that $\|f\|_* = \sup_{Q \in \mathcal{Q}: \|Q\| \leq 1} \langle Q, f \rangle$.

## 2. Preliminaries

The classical statistical learning framework usually considers a dataset $S_n = (Z_1, ..., Z_n)$, made of $n$ i.i.d. elements drawn from a distribution $\mu$ over a measurable instance space $\mathcal{Z}$. Often, one can think of each $Z_i$ as a feature-label pair $(X_i, Y_i)$. Furthermore, we are given a measurable class $\mathcal{W}$ of hypotheses and a loss function $\ell : \mathcal{W} \times \mathcal{Z} \to \mathbb{R}_+$, with $\ell(w, z)$ measuring the quality of the hypothesis $w \in \mathcal{W}$ on the data instance $z \in \mathcal{Z}$. For any given hypothesis $w \in \mathcal{W}$, two key objects of interest are the *training error* $\widehat{\mathcal{L}}(w, S_n) = \frac{1}{n} \sum_{i=1}^{n} \ell(w, Z_i)$ and the *test error* $\mathcal{L}(w) = \mathbb{E}_{Z' \sim \mu}[\ell(w, Z')]$, where the random element $Z'$ has the same distribution as $Z_i$ and is independent of $S_n$. A learning algorithm $\mathcal{A} : \mathcal{Z}^n \to \mathcal{W}$ maps the training sample to an hypothesis in $\mathcal{W}$. More generally, we will focus on randomized learning algorithms, returning a probability distribution $P_{W_n | S_n} \in \Delta_{\mathcal{W}}$ over $\mathcal{W}$, conditionally on $S_n$ (deterministic algorithms can be recovered as special cases, whose the outputs are Dirac distributions). The ultimate goal of the learner is to minimize the test error. Yet, this quantity cannot be computed without knowledge of the data generating distribution $\mu$. In practice, one typically relies on the training error in order to gauge the quality of the algorithm. For an algorithm $\mathcal{A} : S_n \mapsto P_{W_n | S_n}$, we define the *generalization error* as the expected gap between training and test error:

$$\text{Gen}(\mathcal{A}, S_n) = \mathbb{E}\left[ \mathcal{L}(W_n) - \widehat{\mathcal{L}}(W_n, S_n) \Big| S_n \right].$$

The expectation in the above expression integrates over the randomness in the output of the algorithm $W_n \sim P_{W_n | S_n}$, conditionally on the sample $S_n$. We stress that the test error is generally *not* equal to the mean of the training error, due to the dependence of $W_n$ on the training data, which is precisely the challenge that necessitates studying conditions under which the generalization error is small.

We extend the previous setting by considering the case where the data have an intrinsic temporally ordered structure, and come in the form of a stationary process $(Z_t)_{t \in \mathbb{N}^*} \sim \nu$. Formally, we assume that the joint marginal distribution of any block $(Z_t, Z_{t-1}, \ldots, Z_{t-i})$ is the same as the distribution of $(Z_{t+j}, Z_{t+j-1}, \ldots, Z_{t+j-i})$ for any $t$, $i$ and $j$, but the data points are not necessarily independent of each other. In particular, the marginal distribution of $Z_t$ is constant and is denoted by $\mu$. In such a setting, it is natural to continue to use the definition of the test loss and generalization error given above, although with the understanding that $\mu$ now refers to the marginal distribution of an independent copy of $Z_1$, a sample point from a stationary non-i.i.d. process. We remark here that other notions of the test loss may also be considered, and the framework that we propose can be extended to most natural definitions with little work (but potentially large notational overhead). In Section 5, we provide such an extension for a more general setting where the hypotheses themselves are allowed to have memory and the process may not be as strongly stationary as our assumption above requires.

To obtain generalization results we need some control on how strong the dependencies between different datapoints are allowed to be. We hence consider the following assumption.

**Assumption 1** *There exists a non-increasing sequence $(\phi_d)_{d \in \mathbb{N}^*}$ of non-negative real numbers such that, for all $w \in \mathcal{W}$ and all $t \in \mathbb{N}^*$:*

$$\mathbb{E}\left[ \mathcal{L}(w) - \ell(w, Z_t) \Big| \mathcal{F}_{t-d} \right] \leq \phi_d \,,$$

where $\mathcal{L}(w) = \mathbb{E}_{Z' \sim \mu}[\ell(w, Z')]$, *with $Z'$ being independent on the process $(Z_t)_{t \in \mathbb{N}^*}$ and having as distribution the stationary marginal $\mu$ of the $Z_t$.*

The intuition behind this assumption is that the loss associated with the observations $Z_t$ becomes almost independent of the past after $d$ steps, enabling us to treat each sequence of the form $(Z_t, Z_{t+d}, \ldots, Z_{t+(n-t)d})$ as an approximately i.i.d. sequence. Note that this assumption differs from the usual $\beta$-mixing assumption that requires the distribution of $Z_t | \mathcal{F}_{t-d}$ to be close (in total variation) to the marginal distribution $\mu$ for all $t$. Our assumption is somewhat weaker in the sense that it only requires the expected losses under these distributions to be close, and only a one-sided inequality is required. It is easy to verify that our assumption is satisfied if the process is $\beta$-mixing in the usual sense and the losses are bounded in $[0, 1]$.

It is worth noticing that while our assumption can look like a small "cosmetic" improvement over the standard $\beta$-mixing, it can actually be much weaker. For a concrete example, consider let $\ell(w, Z_t) = \ell(w, Z_t') + \varepsilon_t$, where the $Z_t'$ are part of an i.i.d. sequence, and $\varepsilon_t$ is sampled from a bounded $\beta$-mixing process, with $\alpha$ a small constant. Now, for any $w$, $\ell(w, Z_t)$ is clearly a $\beta$-mixing process (inheriting the properties of $\varepsilon_t$), independently of the choice of $\alpha$. In contrast, the mixing rate as defined in our assumption improves linearly with $\alpha$, and vanishes as this parameter approaches zero. As a matter of fact, $\beta$-mixing conditions are very strict in that they require mixing in terms of the entire distribution of the loss, and ignores the "scale" at which non-stationarity impacts the outcomes.

## 3. Proving generalization bounds via online learning

This section introduces our main contribution: a framework for proving generalization bounds for statistical learning on non-i.i.d. data via a reduction to online learning. The framework of online learning focuses on algorithms that aim to improve performance (measured in terms of a given cost function) incrementally as new information becomes available, often without any underlying assumption on how data are generated. The online learner's performance is typically measured in terms of its regret, defined as the the difference between the cumulative cost of the online learner and that of a fixed comparator. We refer to the monographs Cesa-Bianchi and Lugosi (2006) and Orabona (2019) for comprehensive overviews on online learning and regret analysis. Recently, Lugosi and Neu (2023) established a connection between upper bounds on the regret and generalization bounds, showing that the existence of a strategy with a bounded regret in a specially designed online game translates into a generalization bound, via a technique dubbed online-to-PAC conversion. Their focus is on the i.i.d. setting, where the training dataset is made of independent draws. Here, we show that this framework can naturally be extended beyond the i.i.d. assumption.

In what follows, we briefly review the setup of Lugosi and Neu (2023) in Section 3.1 and then describe our new extension of their model to the non-i.i.d. case in Section 3.2. In particular, we prove a high-probability bound for the generalization error of any statistical learning algorithm learnt with a stationary mixing process verifying Assumption 1.

### 3.1. Online-to-PAC conversions for i.i.d. data

The main idea of Lugosi and Neu (2023) is to introduce an online learning *generalization game*, where the following steps are repeated for a sequence of rounds $t = 1, 2, \ldots, n$:

- the online learner picks a distribution $P_t \in \Delta_{\mathcal{W}}$;

- the adversary selects the cost function $c_t : w \mapsto \ell(w, Z_t) - \mathcal{L}(w)$;
- the online learner incurs the cost $\langle P_t, c_t \rangle = \mathbb{E}_{W \sim P_t}[c_t(W)]$;
- $Z_t$ is revealed to the learner.

The learner can adopt any strategy to pick $P_t$, but they can only rely on past knowledge to make their prediction. Explicitly, if $\mathcal{F}_t$ denotes the sigma-algebra generated by $Z_1, ..., Z_t$, then $P_t$ has to be $\mathcal{F}_{t-1}$-measurable. We also emphasize that in this setup the online learner is allowed to know the loss function $\ell$ and the distribution $\mu$ of the data points $Z_t$, and therefore by revealing the value of $Z_t$, the online learner may compute the entire cost function $c_t$.

We define the *regret* of the online learner against the possibly data-dependent *comparator* $P^* \in \Delta_{\mathcal{W}}$ as $\text{Regret}(P^*) = \sum_{t=1}^{n} \langle P_t - P^*, c_t \rangle$. Now, denote as $P_{W_n|S_n}$ the distribution produced by the supervised learning algorithm. With this notation, the generalization error can be written as $\text{Gen}(\mathcal{A}, S_n) = -\frac{1}{n} \sum_{t=1}^{n} \langle P_{W_n|S_n}, c_t \rangle$. By adding and subtracting the quantity $M_n = -\frac{1}{n} \sum_{t=1}^{n} \langle P_t, c_t \rangle$ (corresponding to the total cost incurred by the online learner) we get the following decomposition.

**Theorem 1 (Theorem 1 in Lugosi and Neu, 2023; see appendix A.1)** *With the notation introduced above,*

$$\text{Gen}(\mathcal{A}, S_n) = \frac{\text{Regret}_n(P_{W_n|S_n})}{n} + M_n. \tag{1}$$

The first of these terms correspond to the *regret* of the online learner against a fixed *comparator strategy* that picks $P_{W_n|S_n}$ at each step. The second term is a martingale and can be bounded in high probability with standard concentration tools. Indeed, since $P_t$ is chosen before $Z_t$ is revealed, one can easily check that $\mathbb{E}[\langle P_t, c_t \rangle | \mathcal{F}_{t-1}] = 0$. Thus, to prove a bound on the generalization error of the statistical learning algorithm, it is enough to find an online learning algorithm with bounded regret against $P_{W_n|S_n}$ in the generalization game.

As a concrete application of the above, the following generalization bound is obtained when picking the classic exponential weighted average (EWA) algorithm (Vovk, 1990; Littlestone and Warmuth, 1994; Freund and Schapire, 1997) as our online-learning strategy, and plugging its regret bound into Equation (1).

**Theorem 2 (Corollary 6 in Lugosi and Neu, 2023)** *Suppose that $\ell(w, z) \in [0, 1]$ for all $w, z$. Then, for any $P_1 \in \Delta_{\mathcal{W}}$ and $\eta > 0$, with probability at least $1 - \delta$ on the draw of $S_n$, uniformly on every learning algorithm $\mathcal{A} : S_n \mapsto P_{W_n|S_n}$, we have*

$$\text{Gen}(\mathcal{A}, S_n) \leq \frac{\mathcal{D}_{KL}(P_{W_n|S_n} || P_1)}{\eta n} + \frac{\eta}{2} + \sqrt{\frac{2 \log \left( \frac{1}{\delta} \right)}{n}}.$$

**Proof** We can bound each term of (1) separately. A data-dependent bound for the regret term is obtained via a direct application of the regret analysis of EWA which brings the term $\frac{\mathcal{D}_{KL}(P_{W_n|S_n} || P_1)}{\eta n} + \frac{\eta}{2}$ (see Appendix B.1). The term $\sqrt{\frac{2 \log \left( \frac{1}{\delta} \right)}{n}}$ results from bounding the martingale $M_n$ via an application of Hoeffding–Azuma inequality. ∎

Note that the first term in the bound is data-dependent due to the presence of $P_{W_n|S_n}$, and thus optimizing it requires a data-dependent choice of $\eta$, which is not allowed by Theorem 2. However,

via a union bound argument it is possible to get a bound of the form

$$\text{Gen}(\mathcal{A}, S_n) = \mathcal{O}\left(\sqrt{\frac{\mathcal{D}_{KL}(P_{W_n|S_n}||P_1)}{n}} + \sqrt{\frac{1}{n}\log\left(\frac{\log n}{\delta}\right)}\right),$$

For the details, we refer to the proof of Corollary 5 of Lugosi and Neu (2023), which recovers a classical PAC-Bayes bound of McAllester (1998).

### 3.2. Online-to-PAC conversions for non-i.i.d. data

We will now drop the i.i.d. condition, and instead consider non-i.i.d. data sequences satisfying Assumption 1. For this setting we define the following variant of the generalization game.

**Definition 3 (Generalization game with delay)** *The generalization game with delay $d \in \mathbb{N}^*$ is an online learning game where the following steps are repeated for a sequence of rounds $t = 1, ..., n$:*

- *the online learner picks a distribution $P_t \in \Delta_{\mathcal{W}}$;*
- *the adversary selects the cost function $c_t : w \mapsto \ell(w, Z_t) - \mathcal{L}(w)$;*
- *the online learner incurs the cost $\langle P_t, c_t \rangle = \mathbb{E}_{W \sim P_t}[c_t(W)]$;*
- *if $t \geq d$, $Z_{t-d+1}$ is revealed to the learner.*

As before, the last step effectively reveals the entire cost function $c_{t-d+1}$ to the online learner. The main difference between our version of the generalization game and the standard one of Lugosi and Neu (2023) is the introduction of a *delay* on the online learning algorithm's decisions. Specifically, we will force the online learner to only take information into account up to time $t - d$ when picking their action $P_t$. Clearly, setting $d = 1$ recovers the original version of the generalization game.

It is easy to see that the regret decomposition of Theorem 1 still remains valid in the current setting. The purpose of introducing the delay is to be able to make sure that the term $M_n = -\frac{1}{n}\sum_{t=1}^n \langle P_t, c_t \rangle$ is small. The lemma below states that the increments of $M_n$ behave similarly to a martingale-difference sequence, thanks to the introduction of the delay.

**Lemma 4** *Fix $d \in [\![1, n]\!]$. Under Assumption 1, defining $P_t$ and $c_t$ as in 3, for all $t \in [\![1, n]\!]$*

$$\mathbb{E}[\langle -P_t, c_t \rangle | \mathcal{F}_{t-d}] \leq \phi_d.$$

**Proof** Since $P_t$ is $\mathcal{F}_{t-d}$-measurable we have $\mathbb{E}[\langle -P_t, c_t \rangle | \mathcal{F}_{t-d}] = \langle P_t, \mathbb{E}[-c_t | \mathcal{F}_{t-d}] \rangle \leq \phi_d$, where the last step uses Assumption 1. ■

Thus, by following the decomposition of Theorem 1, we are left with the problem of bounding the regret of the delayed online learning algorithm against $P_{W_n|S_n}$, denoted as $\text{Regret}_{d,n}(P_{W_n|S_n}) = \sum_{t=1}^n \langle P_t - P_{W_n|S_n}, c_t \rangle$. The following proposition states a simple and clean bound that one can immediately derive from these insights.

**Proposition 5 (Bound in expectation)** *Consider $(Z_t)_{t \in \mathbb{N}^*}$ satisfying Assumption 1 and suppose there exists a $d$-delayed online learning algorithm with regret bounded by $\text{Regret}_{d,n}(P^*)$ against any comparator $P^*$. Then, the expected generalization of $\mathcal{A}$ is bounded as*

$$\mathbb{E}\left[\text{Gen}(\mathcal{A}, S_n)\right] \leq \frac{\mathbb{E}\left[\text{Regret}_{d,n}(P_{W_n|S_n})\right]}{n} + \phi_d.$$

**Proof** By Theorem 1, it holds that $\mathbb{E}[\text{Gen}(\mathcal{A}, S_n)] = \frac{\mathbb{E}[\text{Regret}_{d,n}(P_{W_n|S_n})]}{n} + \mathbb{E}[M_n]$, where the regret is for a strategy $P_t$ in the delayed generalization game. Hence, by Lemma 4

$$\mathbb{E}[M_n] = \mathbb{E}\left[-\frac{1}{n}\sum_{t=1}^{n}\langle P_t, c_t\rangle\right] = \frac{1}{n}\sum_{t=1}^{n}\mathbb{E}[\langle -P_t, c_t\rangle] = \frac{1}{n}\sum_{t=1}^{n}\mathbb{E}\left[\mathbb{E}[\langle -P_t, c_t\rangle|\mathcal{F}_{t-d}]\right] \leq \phi_d,$$

which proves the claim. ∎

The above result holds in expectation over the training sample. We now provide a high-probability guarantee on the generalization error.

**Theorem 6 (Bound in probability)** *Assume that $(Z_t)_{t\in\mathbb{N}^*}$ satisfies Assumption 1 and consider a d-delayed online learning algorithm with regret bounded by $\text{Regret}_{d,n}(P^*)$ against any comparator $P^*$. Then, for any $\delta > 0$, it holds with probability $1 - \delta$ on the draw of $S_n$, uniformly for all $\mathcal{A}$,*

$$\text{Gen}(\mathcal{A}, S_n) \leq \frac{\text{Regret}_{d,n}(P_{W_n|S_n})}{n} + \phi_d + \sqrt{\frac{2d\log\left(\frac{1}{\delta}\right)}{n}}.$$

The proof of this claim follows directly from combining the decomposition of Theorem 1 with a standard concentration result for mixing processes that we state below.

**Lemma 7** *Fix $d \in [\![1, n]\!]$ and consider $(Z_t)_{t\in\mathbb{N}^*}$ satisfying Assumption 1. Consider the generalization game of Definition 3. Then, for any $\delta > 0$, the following bound is satisfied with probability at least $1 - \delta$:*

$$M_n \leq \phi_d + \sqrt{\frac{2d\log\left(\frac{1}{\delta}\right)}{n}}.$$

The proof is based on a classic "blocking" technique due to Yu (1994). For the sake of completeness, we provide a proof in Appendix A.2.

## 4. New generalization bounds for non-i.i.d. data

The dependence on the delay $d$ for the bounds that we presented in the previous section is non-trivial. Indeed, if on the one hand increasing the delay will reduce the magnitude of $\phi_d$, on the other hand the regret of the online learner will grow with $d$. There is hence a trade-off between these two terms appearing in our bounds. In what follows, we derive some concrete generalization bounds from Theorem 6, under a number of different choices of the online learning algorithm. For concreteness, we will consider two types of mixing assumptions, but stress that the approach can be applied to any process that satisfies Assumption 1.

### 4.1. Regret bounds for delayed online learning

From Theorem 6, we can obtain a generalization bound using our framework if we have a regret bound for a delayed online algorithm. This is a well-known problem in the area of online learning (see, e.g., Weinberger and Ordentlich, 2002; Joulani et al., 2013). In the following, we will leverage the following simple trick that allows us to extend the regret bounds of any online learning algorithm to its delayed counterpart, provided that the regret bound respects some specific assumptions.

**Lemma 8 ([Weinberger and Ordentlich, 2002](#))** *Consider any online algorithm whose regret satisfies* $\mathrm{Regret}_n(P^*) \leq R(n)$ *for any comparator* $P^*$, *where* $R$ *is a non-decreasing real-valued function such that* $y \mapsto yR(x/y)$ *is concave in* $y$ *for any fixed* $x$. *Then, for any* $d \geq 1$ *there exists an online learning algorithm with delay* $d$ *such that, for any* $P^*$,

$$\mathrm{Regret}_{d,n}(P^*) \leq dR\left(n/d\right) .$$

The proof idea is closely related to the blocking trick of [Yu](#) ([1994](#)), with an algorithmic construction that runs one instance of the base method for each index $i = 1, 2, \ldots, d$, with the $i$-th instance being responsible for the regret in rounds $i, i + d, i + 2d, \ldots$ (more details are provided in Appendix [B.3](#)). For most of the regret bounds that we consider, the function $R$ takes the form $R(n) = O(\sqrt{n})$, so that the first term in the generalization bound is typically of order $\sqrt{d/n}$. Since this term matches the bound on $M_n$ in Lemma [7](#), in this case the final generalization bound behaves effectively as if the sample size was $n/d$ instead of $n$.

We remark that that the regret guarantees of [Weinberger and Ordentlich](#) ([2002](#)) are minimax optimal ([Joulani et al., 2013](#))[1] , and although more efficient methods exist for online learning with delays, the online-to-PAC reduction does not require the strategy to be executed in practice, making the computational efficiency of the online method irrelevant to our analysis.

### 4.2. Geometric and algebraic mixing

The following definition gives two concrete examples of mixing processes that satisfy Assumption [1](#) with different choices of $\phi_d$, and are commonly considered in the related literature (see, e.g., [Mohri and Rostamizadeh, 2010](#), [Levin and Peres, 2017](#)).

**Definition 9** *We say that a stationary process* $(Z_t)_{t \in \mathbb{N}^*}$ *satisfying Assumption [1](#) is* geometrically mixing *if* $\phi_d = Ce^{-\frac{d}{\tau}}$, *for some positive* $\tau$ *and* $C$, *and* algebraically mixing *if* $\phi_d = Cd^{-r}$, *for some positive* $r$ *and* $C$.

Instantiating the bound of Theorem [6](#) to these two cases yields the following two corollaries.

**Corollary 10** *Assume* $(Z_t)_{t \in \mathbb{N}^*}$ *is a geometrically mixing process with constants* $\tau, C > 0$. *Consider a* $d$-*delayed online learning algorithm with regret bounded by* $\mathrm{Regret}_{d,n}(P^*)$ *for all comparators* $P^*$. *Then, setting* $d = \lceil \tau \log n \rceil$, *for any* $\delta > 0$, *with probability at least* $1 - \delta$ *we have that, uniformly for any algorithm* $\mathcal{A}$,

$$\mathrm{Gen}(\mathcal{A}, S_n) \leq \frac{\mathrm{Regret}_{d,n}(P_{W_n|S_n})}{n} + \frac{C}{n} + \sqrt{\frac{2\left(\tau \log n + 1\right) \log\left(\frac{1}{\delta}\right)}{n}} .$$

Up to a term linear in $\tau$ and some logarithmic factors, the above states that under the geometric mixing the same rates are achievable as in the i.i.d. setting. Roughly speaking, this amounts to saying that the effective sample size is a factor $\tau$ smaller than the original number of samples $n$, as long as generalization is concerned.

---

1. The guarantees of Lemma [8](#) are minimax optimal in the sense that the best that one can hope for without further assumptions on the loss is a regret of $O(\sqrt{dT})$. Although it is true that better regret bounds can be achieved under more specific assumptions about the losses (see Table 1 in [Joulani et al., 2013](#)), this is precisely the type of assumption that we wanted to avoid in our work.

**Corollary 11** *Assume* $(Z_t)_{t \in \mathbb{N}^*}$ *is an algebraic mixing process with constants* $r, C > 0$. *Consider a* $d$-*delayed online learning algorithm with regret bounded by* $\text{Regret}_{d,n}(P^*)$ *against any comparator* $P^*$. *Then, setting* $d = (C^2 n)^{1/(1+2r)}$, *for any* $\delta > 0$, *with probability at least* $1 - \delta$ *we have that, uniformly for any algorithm* $\mathcal{A}$,

$$\text{Gen}(\mathcal{A}, S_n) \leq \frac{\text{Regret}_{d,n}(P_{W_n|S_n})}{n} + C\left(1 + \sqrt{\log(1/\delta)}\right) n^{-\frac{2r}{2(1+2r)}}.$$

This result suggests that the rates achievable for algebraically mixing processes are qualitatively much slower than what one can get for i.i.d. or geometrically mixing data sequences (although the rates do eventually approach $1/\sqrt{n}$ as $r$ goes to infinity).

### 4.3. Multiplicative weights with delay

We now turn our attention to picking online strategies for the purpose of bounding the main term in the decomposition of the generalization error. We start by focusing on the classic exponential weighted average (EWA) algorithm (Vovk, 1990; Littlestone and Warmuth, 1994; Freund and Schapire, 1997). We fix a data-free prior $P_1 \in \Delta_{\mathcal{W}}$ and a learning rate parameter $\eta > 0$. We consider the updates

$$P_{t+1} = \arg\min_{P \in \Delta_{\mathcal{W}}} \left\{ \langle P, c_t \rangle + \frac{1}{\eta} \mathcal{D}_{KL}(P||P_t) \right\},$$

Combining the standard regret bound of EWA (see Appendix B.1) with Lemma 8 and Corollary 10 yields the result that follows.

**Corollary 12** *Suppose that* $(Z_t)_{t \in \mathbb{N}^*}$ *is a geometric mixing process with constants* $\tau, C > 0$. *Suppose that* $\ell(w, z) \in [0, 1]$ *for all* $w, z$. *Then, for any* $P_1 \in \Delta_{\mathcal{W}}$ *and any* $\delta > 0$, *with probability at least* $1 - \delta$, *uniformly on any learning algorithm* $\mathcal{A}$ *we have*

$$\text{Gen}(\mathcal{A}, S_n) \leq \frac{\mathcal{D}_{KL}(P_{W_n|S_n}||P_1)(\tau \log n + 1)}{\eta n} + \frac{\eta}{2} + \frac{C}{n} + \sqrt{\frac{2(\tau \log n + 1)\log\left(\frac{1}{\delta}\right)}{n}}.$$

This results suggests that when considering geometric mixing processes, by applying a union bound over a well-chosen range of $\eta$ we recover the PAC-Bayes bound of McAllester (1998) up to a $O(\sqrt{\tau \log n})$ factor that quantifies the price of dropping the i.i.d. assumption. A similar result can be derived from Corollary 11 for algebraically mixing processes, leading to a bound typically scaling as $n^{-2r/(2(1+2r))}$.

### 4.4. Follow the regularized leader with delay

We finally extend the common class of online learning algorithms known as *follow the regularized leader* (FTRL, see, e.g., Abernethy and Rakhlin, 2009; Orabona, 2019) to the problem of learning with delay. FTRL algorithms are defined using a convex regularization function $h : \Delta_{\mathcal{W}} \to \mathbb{R}$. We restrict ourselves to the set of proper, lower semi-continuous and $\alpha$-strongly convex functions with respect to a norm $\|\cdot\|$ (and its respective dual norm $\|\cdot\|_*$) defined on the set of signed finite measures on $\mathcal{W}$ (see Appendix B.2 for more details). The updates of of the FTRL algorithm (without delay) are defined as follows:

$$P_{t+1} = \arg\min_{P \in \Delta_{\mathcal{W}}} \left\{ \sum_{s=1}^{t} \langle P, c_s \rangle + \frac{1}{\eta} h(P) \right\}.$$

The existence of the minimum is guaranteed by the compactness of $\Delta_{\mathcal{W}}$ under $\|\cdot\|$, and its uniqueness is ensured by the strong convexity of $h$. Combining the analysis of FTRL (see Appendix B.2) with Lemma 8 and Corollary 10 yields the following result.

**Corollary 13** *Suppose that $(Z_t)_{t\in\mathbb{N}^*}$ is a geometric mixing process with constants $\tau, C > 0$. Suppose that $\ell(w,z) \in [0,1]$ for all $w, z$. Assume there exists $B > 0$ such that for all $t, \|c_t\|_* \leq B$. Then, for any $P_1 \in \Delta_{\mathcal{W}}$, for any $\delta > 0$ with probability at least $1 - \delta$ on the draw of $S_n$, uniformly for all $\mathcal{A}$,*

$$\mathrm{Gen}(\mathcal{A}, S_n) \leq \frac{\left(h(P_{W_n|S_n}) - h(P_1)\right)(\tau \log n + 1)}{\eta n} + \frac{\eta B^2}{2\alpha} + \frac{C}{n} + \sqrt{\frac{2(\tau \log n + 1)\log\left(\frac{1}{\delta}\right)}{n}}.$$

This generalization bound is similar to the bound of Theorem 9 of Lugosi and Neu (2023) up to a $O(\sqrt{\tau \log n})$ factor, when applying a union-bound argument over an appropriate grid of learning-rates $\eta$. In particular, this result recovers PAC-Bayesian bounds like those of Corollary 12 when choosing $h = \mathcal{D}_{\mathrm{KL}}(\cdot\|P_1)$. We refer to Section 3.2 in Lugosi and Neu (2023) for more discussion on such bounds. As before, a similar result can be stated for algebraically mixing processes, with the main terms scaling as $n^{-2r/2(1+2r)}$ instead of $n^{-1/2}$.

## 5. Generalization bounds for dynamic hypotheses

Finally, inspired by the works of Eringis et al. (2022, 2024), we extend our framework to accommodate loss functions $\ell$ that rely not only on the last data point $Z_t$, but on the entire data sequence $\overline{Z}_t = (Z_t, Z_{t-1}, \ldots, Z_1)$. Formally, we will consider loss functions of the form $\ell : \mathcal{W} \times \mathcal{Z}^* \to \mathbb{R}_+$[2] and write $\ell(w, \overline{z}_t)$ to denote the loss associated with hypothesis $w \in \mathcal{W}$ on sequence $\overline{z}_t \in \mathcal{Z}^t$. This consideration extends the learning problem to class of dynamical predictors such as Kalman filters, autoregressive models, or recurrent neural networks (RNNs), broadly used in time-series forecasting (Ariyo et al., 2014; Takeda et al., 2016). Specifically, if we think of $z_t = (x_t, y_t)$ as a data-pair of context and observation, in time-series prediction we usually not only rely on the context $x_t$ but also on the past sequence of contexts and observations $(x_{t-1}, y_{t-1}, \ldots, x_1, y_1)$. As an example, consider $\ell(w, z_t, \ldots, z_1) = \frac{1}{2}(y_t - h_w(x_t, z_{t-1}, \ldots, z_1))^2$ where $h \in \mathcal{H}$ is a function class parameterized by $\mathcal{W}$. For this type of loss function a natural definition of the test error is:

$$\widetilde{\mathcal{L}}(w) = \lim_{n\to\infty} \mathbb{E}[\ell(w, Z'_t, Z'_{t-1}, \ldots, Z'_{t-n})],$$

where $\overline{Z}'_t = (Z'_t, Z'_{t-1}, \ldots)$ is a semi-infinite random sequence drawn from the same stationary process that has generated the data $\overline{Z}_t$. We consider the following assumption.

**Assumption 2** *For a given process $(Z_t)_{t\in\mathbb{Z}}$ with joint-distribution $\nu$ over $\mathcal{Z}^{\mathbb{Z}}$ and same marginals $\mu$ over $\mathcal{Z}$, there exists a non-increasing sequence $(\phi_d)_{d\in\mathbb{N}^*}$ of non-negative real numbers such that the following holds for all $w \in \mathcal{W}$, for all $t \in \mathbb{N}^*$:*

$$\mathbb{E}\left[\ell(w, Z_t, \ldots, Z_1) - \widetilde{\mathcal{L}}(w)\Big|\mathcal{F}_{t-d}\right] \leq \phi_d.$$

---

2. Here, $\mathcal{Z}^*$ denotes the disjoint union $\mathcal{Z}^* = \sqcup_{t\in\mathbb{N}}\mathcal{Z}^t$.

This is a generalization of Assumption 1 in the sense that taking $\ell(w, Z_t, \ldots, Z_1) = \ell(w, Z_t)$ simply amounts to requiring the same mixing condition as before. For our online-to-PAC conversion we consider the same framework as in Definition 3, except that now the cost function is defined as

$$c_t : w \mapsto \ell(w, Z_t, \ldots, Z_1) - \widetilde{\mathcal{L}}(w).$$

It easy to check that result of Lemma 7 still holds for this specific cost, and we can thus extend all the results of Section 4. We state the following adaptation of Theorem 6 below.

**Theorem 14** *Assume $(Z_t)_{t\in\mathbb{Z}}$ which satisfies Assumption 2 and consider a $d$-delayed online learning algorithm with regret bounded by $\mathrm{Regret}_{d,n}(P^*)$ against any comparator $P^*$. Then, for any $\delta > 0$, it holds with probability $1 - \delta$:*

$$\mathrm{Gen}(\mathcal{A}, S_n) \leq \frac{\mathrm{Regret}_{d,n}(P_{W_n|S_n})}{n} + \phi_d + \sqrt{\frac{2d\log\left(\frac{1}{\delta}\right)}{n}}.$$

To see that Assumption 2 can be verified and the resulting bounds can be meaningfully applied, consider the following concrete assumptions about the hypothesis class, the loss function, and the data generating process. The first assumption says that for any given hypothesis, the influence of past data points on the associated loss vanishes with time (*i.e.*, the hypothesis forgets the old data points at a controlled rate).

**Assumption 3** *There exists a decreasing non-negative sequence $(B_d)_{d\in\mathbb{N}^*}$ such that, for any two sequences $\overline{z}_t = (z_t, \ldots, z_i)$ and $\overline{z}'_t = (z'_t, \ldots, z'_j)$ of possibly different lengths that satisfy $z_k = z'_k$ for all $k \in t, \ldots, t - d + 1$, we have $|\ell(w, \overline{z}_t) - \ell(w, \overline{z}'_t)| \leq B_d$, for all $w \in \mathcal{W}$.*

This condition can be verified for stable dynamical systems like autoregressive models, certain classes of RNNs, or sequential predictors that have bounded memory by design (see Eringis et al., 2022, 2024). The next assumption is a refinement of Assumption 1, adapted to the case where the loss function acts on blocks of $d$ data points $\overline{z}_{t-d+1:t} = (z_t, z_{t-1}, \ldots, z_{t-d+1})$.

**Assumption 4** *Let $\overline{Z}_t = (Z_t, \ldots, Z_1)$ be a sequence of data points and $\overline{Z}'_t = (Z'_t, \ldots, Z'_0, \ldots)$ an independent copy of the same process. Then, there exists a decreasing sequence $(\beta_d)_{d\in\mathbb{N}^*}$ non-negative real numbers such that the following is satisfied for all hypotheses $w \in \mathcal{W}$ and all $d \in \mathbb{N}^*$:*

$$\mathbb{E}\left[\ell(w, \overline{Z}'_{t-d+1:t}) - \ell(w, \overline{Z}_{t-d+1:t}) \,\middle|\, \mathcal{F}_{t-2d}\right] \leq \beta_d.$$

This assumption can be verified whenever the loss function is bounded and the joint distribution of the data block $\overline{Z}_{t-d+1:t}$ satisfies a $\beta$-mixing assumption. This latter condition amounts to requiring that the conditional distribution of each data block given a block that trails $d$ steps behind is close to the marginal distribution in total variation distance, up to an additive term of $\beta_d$. The next proposition shows that these two simple conditions together imply that Assumption 2 holds, and the bound of Theorem 14 can be meaningfully instantiated for bounded-memory hypothesis classes deployed on mixing processes.

**Proposition 15** *Suppose that the loss function satisfies Assumption 3 and the data distribution satisfies Assumption 4. Then Assumption 2 is satisfied with $\phi_d = 2B_{d/2} + \beta_{d/2}$.*

## 6. Conclusion

We have developed a general framework for deriving generalization bounds for non-i.i.d. processes under a general mixing assumption, extending the online-to-PAC-conversion framework of Lugosi and Neu (2023). Among other results, this approach has allowed us to prove PAC-Bayesian generalization bounds for such data in a clean and transparent way, and even study classes of dynamic hypotheses under a simple bounded-memory condition. We now conclude by mentioning links with the most closely related previous works in the literature.

The PAC-Bayesian bound in Corollary 12 is closely related to Theorem 19 of Ralaivola et al. (2010). Their bound has the advantage to upper-bound a nonlinear proxy of the generalization error and may imply faster rates if the training error is close to zero. Conversely, the proof of this result is a rather cumbersome application of the heavy machinery developed in the rest of their paper. In comparison, our proofs are direct and straightforward, and generalize easily to other divergences than the relative entropy, as discussed in Section 4.4.

Our results concerning dynamic hypotheses provide a clean and tight alternative to the results of (Alquier and Wintenberger, 2012; Eringis et al., 2022). Our Assumptions 3 and 4 can be both shown to hold under their conditions, and the dependence of our bounds on the parameters appearing in these assumptions are stated explicitly. This is a significant advantage over the guarantees of Eringis et al. (2022), which are stated without clearly spelling out the dependence of the problem parameters they consider. Finally, their bounds are stated in PAC-Bayesian terms, which is just one instance of the variety of generalization bounds that our framework can recover. It is worth mentioning that there are other models for studying non-stationarity in statistical learning (León and Perron, 2004; Simchowitz et al., 2018). While our techniques may not be directly applicable to these settings, we believe that a similar online-learning-based perspective might prove helpful for these purposes too.

We also wish to comment on the concurrent work of Chatterjee et al. (2025) who recently proposed a very similar framework for extending the Online-to-PAC technique of Lugosi and Neu (2023) to deal with non-i.i.d. data. Their assumptions and techniques are closely related to ours, except for the major difference that their reduction does not involve delays. Instead, Chatterjee et al. (2025) make strong assumptions about both the loss function and the online learning algorithm: the losses are assumed to be Lipschitz and the online learning method to be "stable" in a rather strong sense. These assumptions are inherited from Agarwal and Duchi (2012) who made the same type of assumptions in the context of proving excess risk bounds for online-to-batch conversions in non-i.i.d. settings. Thanks to the use of delays, our framework does not require any such assumptions, and yields strictly tighter bounds. We note that this observation can be easily used to improve the guarantees of Agarwal and Duchi (2012): by introducing delays into their algorithmic reduction, one can remove the stability conditions required for their results.

Given these positive results, we believe that our results further demonstrate the power of the Online-to-PAC framework of Lugosi and Neu (2023), and clearly show that it is particularly promising for developing techniques for generalization in non-i.i.d. settings. We hope that the flexibility of our framework will find further uses and enable more rapid progress in the area.

## Acknowledgments

This project has received funding from the European Research Council (ERC) under the European Union's Horizon 2020 research and innovation programme (Grant agreement No. 950180). The

authors would like to thank Yoav Freund for valuable and insightful discussions during the preparation of this work, and the anonymous reviewers for their constructive feedback.

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

## Appendix A. Omitted proofs

### A.1. The proof of Theorem 1

Let $(P_t)_{t=1}^n \in \Delta_{\mathcal{W}}^n$ be the predictions of an online learner playing the *generalization game*. Then

$$
\begin{aligned}
\operatorname{Gen}(\mathcal{A}, S_n) &= \frac{1}{n} \sum_{t=1}^n \mathbb{E}[\ell_t(W_n) - \mathcal{L}(W_n)|S_n] \\
&= -\frac{1}{n} \sum_{t=1}^n \mathbb{E}[c_t(W_n)|S_n] \\
&= -\frac{1}{n} \sum_{t=1}^n \langle P_{W_n|S_n}, c_t \rangle \\
&= \frac{1}{n} \sum_{t=1}^n \langle P_t - P_{W_n|S_n}, c_t \rangle - \frac{1}{n} \sum_{t=1}^n \langle P_t, c_t \rangle \\
&= \frac{\operatorname{Regret}_n(P_{W_n|S_n})}{n} + M_n.
\end{aligned}
$$

### A.2. Proof of lemma 7

Assume $n = Kd$ and denote $M_n = \sum_{t=1}^n \langle P_t, c_t \rangle$. Then

$$
M_n = \sum_{i=1}^d \sum_{t=1}^K \langle P_{i+d(t-1)}, c_{i+d(t-1)} \rangle.
$$

We denote $X_t^{(i)} = \langle P_{i+d(t-1)}, c_{i+d(t-1)} \rangle$ and we want to bound in high-probability the term $M_n = \sum_{i=1}^d M_i$, where $M_i = \sum_{t=1}^K X_t^{(i)}$. We notice that $f \mapsto \log \mathbb{E}\left[e^{\lambda f}\right]$ is convex. Therefore, from Jensen's inequality, for any positive $p_1, \ldots, p_d$ such that $\sum_{i=1}^M p_i = 1$, and for any $\lambda > 0$, it holds that

$$
\log \mathbb{E}\left[e^{\lambda M_n}\right] \leq \sum_{i=1}^d p_i \log \mathbb{E}\left[e^{\lambda \frac{M_i}{p_i}}\right].
$$

Let us denote $\mathcal{F}_t^{(i)} = \mathcal{F}_{i+d(t-1)}$, we have for all $i \in [d]$

$$
\mathbb{E}\left[e^{\lambda \frac{M_i}{p_i}}\right] = \mathbb{E}\left[e^{\frac{\lambda}{p_i} \sum_{t=1}^K X_t^{(i)}}\right] = \mathbb{E}\left[e^{\frac{\lambda}{p_i} \sum_{t=1}^{K-1} X_t^{(i)}} \mathbb{E}\left[e^{\frac{\lambda}{p_i} X_K^{(i)}} \Big| \mathcal{F}_{K-1}^{(i)}\right]\right].
$$

Now remark that

$$
\mathbb{E}\left[e^{\frac{\lambda}{p_i} X_K^{(i)}} \Big| \mathcal{F}_{K-1}^{(i)}\right] = \mathbb{E}\left[e^{\frac{\lambda}{p_i}(X_K^{(i)} - \mathbb{E}[X_K^{(i)}|F_{K-1}^{(i)}])} \Big| F_{K-1}^{(i)}\right] e^{\frac{\lambda}{p_i} \mathbb{E}[X_K^{(i)}|F_{K-1}^{(i)}]}.
$$

Denote $Z = X_K^{(i)} - \mathbb{E}[X_K^{(i)}|F_{K-1}^{(i)}]$. Note that $|Z| \leq 2$ and $\mathbb{E}[Z|F_{K-1}^{(i)}] = 0$. By Hoeffding's inequality we have $\mathbb{E}\left[e^{\frac{\lambda}{p_i} X_K^{(i)}} \Big| \mathcal{F}_{K-1}^{(i)}\right] \leq e^{\frac{\lambda^2}{2p_i^2} + \frac{\lambda \phi_d}{p_i}}$. Repeating this reasoning $K$ times yields

$$
\mathbb{E}\left[e^{\lambda \frac{M_i}{p_i}}\right] \leq e^{\frac{\lambda^2 K}{2p_i^2} + \frac{\lambda K \phi_d}{p_i}}.
$$

Finally,

$$\log \mathbb{E}\left[e^{\lambda M_n}\right] \leq \sum_{i=1}^{d} p_i \left( \frac{\lambda^2 K}{2p_i^2} + \frac{\lambda K \phi_d}{p_i} \right)$$

now taking $p_i = \frac{1}{d}$ essentially gives $\log \mathbb{E}\left[e^{\lambda M_n}\right] \leq n(\frac{\lambda^2 d}{2} + \lambda \phi_d)$. Now, from Chernoff's inequality

$$\mathbb{P}\left( \frac{M_n}{n} \geq t \right) \leq \mathbb{E}\left[e^{\lambda \frac{M_n}{n}}\right] e^{-\lambda t} \leq e^{\frac{\lambda^2 d}{2n} + \lambda \phi_d - \lambda t} \leq e^{-\frac{n(t-\phi_d)^2}{2d}}.$$

Setting $t = \phi_d + \sqrt{\frac{2d \log(\frac{1}{\delta})}{n}}$ concludes the proof.

### A.3. Proof of Proposition 15

Suppose without loss of generality that $d$ is even and define $d' = d/2$. For the proof, let $\overline{Z}'_n$ be a semi-infinite sequence drawn independently from the same process as $\overline{Z}_n$. Then, we have

$$\begin{aligned}
\tilde{\mathcal{L}}(w) &= \lim_{n \to \infty} \mathbb{E}[\ell(w, Z'_t, Z'_{t-1}, ..., Z'_{t-n})] \\
&\leq \mathbb{E}[\ell(w, Z'_t, Z'_{t-1}, \ldots, Z'_{t-d'})] + B_{d'} \\
&\leq \mathbb{E}\left[\ell(w, Z_t, Z_{t-1}, \ldots, Z_{t-d'}) \mid \mathcal{F}_{t-2d'}\right] + B_{d'} + \beta_{d'} \\
&\leq \mathbb{E}\left[\ell(w, Z_t, Z_{t-1}, \ldots, Z_{t-d'}, \ldots, Z_1) \mid \mathcal{F}_{t-2d'}\right] + 2B_{d'} + \beta_{d'} \\
&\leq \mathbb{E}\left[\ell(w, Z_t, Z_{t-1}, \ldots, Z_1) \mid \mathcal{F}_{t-2d'}\right] + 2B_{d'} + \beta_{d'},
\end{aligned}$$

where we used Assumption 3 in the first inequality, Assumption 4 in the second one, and Assumption 3 again in the last step. This proves the statement.

## Appendix B. Online Learning Tools and Results

### B.1. Regret Bound for EWA

Recalling EWA updates we have:

$$P_{t+1} = \arg \min_{P \in \Delta_{\mathcal{W}}} \left\{ \langle P, c_t \rangle + \frac{1}{\eta} \mathcal{D}_{KL}(P \| P_t) \right\},$$

where $\eta > 0$ is a learning-rate parameter. The minimizer can be shown to exist and satisfies:

$$\frac{\mathrm{d}P_{t+1}}{\mathrm{d}P_t}(w) = \frac{e^{-\eta c_t(w)}}{\int_{\mathcal{W}} e^{-\eta c_t(w')} \mathrm{d}P_t(w')},$$

and the following result holds.

**Proposition 16** *For any prior $P_1 \in \Delta_{\mathcal{W}}$ and any comparator $P^* \in \Delta_{\mathcal{W}}$ the regret of EWA simultaneously satisfies for $\eta > 0$:*

$$\mathrm{Regret}(P^*) \leq \frac{\mathcal{D}_{KL}(P^* \| P_1)}{\eta} + \frac{\eta}{2} \sum_{t=1}^{n} \|c_t\|_\infty^2.$$

We refer the reader to Appendix A.1 of Lugosi and Neu (2023) for a complete proof of the result above.

### B.2. Regret Bound for FTRL

We say that $h$ is $\alpha-$strongly convex if the following inequality is satisfied for all $P, P' \in \Delta_{\mathcal{W}}$ and all $\lambda \in [0, 1]$:

$$h(\lambda P + (1 - \lambda)P') \leq \lambda h(P) + (1 - \lambda)h(P') - \frac{\alpha\lambda(1 - \lambda)}{2}||P - P'||^2.$$

Recalling the FTRL updates:

$$P_{t+1} = \underset{P \in \Delta_{\mathcal{W}}}{\arg\min} \left\{ \sum_{s=1}^{t} \langle P, c_s \rangle + \frac{1}{\eta} h(P) \right\},$$

the following results holds.

**Proposition 17** *For any prior $P_1 \in \Delta_{\mathcal{W}}$ and any comparator $P^* \in \Delta_{\mathcal{W}}$ the regret of FTRL simultaneously satisfies for $\eta > 0$:*

$$\text{Regret}_n(P^*) \leq \frac{h(P^*) - h(P_1)}{\eta} + \frac{\eta}{2\alpha} \sum_{t=1}^{n} ||c_t||_*^2.$$

We refer the reader to Appendix A.3 of Lugosi and Neu (2023) for a complete proof of the results above.

### B.3. Details about the reduction of Weinberger and Ordentlich (2002)

For concretenes we formally present how to turn any online learning algorithm into its delayed version. For sake of convenience, assume $n = Kd$. We denote $\tilde{c}_t^{(i)} = c_{i+d(t-1)}$ (for instance $\tilde{c}_1^{(1)} = c_1$ is the cost revealed at time $d + 1$). Then we create $d$ instances of horizon time $K$ of the online learning as follows, for $i = 1, \ldots, d$:

- We initialize $\tilde{P}_1^{(i)} = P_0$,
- for each block $i$ of length $K$ we update for $t = 1, \ldots, K$:

$$\tilde{P}_{t+1}^{(i)} = \text{OL}_{\text{update}}\left( (\tilde{c}_s^{(i)})_{s=1}^t \right).$$

Here $\text{OL}_{\text{update}}$ refers to the update function of the online learning algorithm we consider which can possibly depend of the whole history of cost functions (e.g., in the case of the FTRL update).

