# OpenReview forum: "Generalization bounds for mixing processes via delayed online-to-PAC conversions"
_algorithmiclearningtheory.org/ALT/2025/Conference — ALT 2025_

### Official Review · Reviewer_ocxV · 2024-11-08
**Comments on Paper 101**

**Rating:** 7
**Confidence:** 3

**Review:**

#Summary

This paper studies the generalization error of statistical learning algorithms in a non i.i.d. setting, where the training data is sampled from a stationary mixing process. Specifically, the authors propose a novel analytic framework, which formulates an online generalization game with delayed feedback and can achieve a reduction from the regret of online learning with delayed feedback into the generalization error bound. Based on this framework, two concrete examples of mixing processes (i.e., the geometric mixing and the algebraically mixing) are considered, yielding the corresponding generalization error bounds. Moreover, the authors also extend their framework into the case that loss functions rely not only on the last data, but on all historical data.

#Strength
1) The non i.i.d setting studied in this paper may be more practical than the classical i.i.d setting. Although this paper is not the first to investigate the generalization error in the non i.i.d setting, the assumption made in this paper seems to be weaker than that in previous studies.
2) The analytic framework is proposed by establishing the connection between the regret of online learning with delayed feedback and the generalization error, which is new and interesting (at least for me).
3) Based on the analytic framework, the authors have recovered near-optimal rates in a number of well-studied settings by tuning the delay appropriately.

#Weakness or Questions
1) It seems that the main results of the analytic framework (i.e., Proposition 5 and Theorem 6) can be simply proved by combining the Theorem 1 in Lugosi and Neu (2023) with Assumption 1 in this paper.
2) According to the online generalization game with delayed feedback formulated in this paper, the learner needs to know the cost function $c_t$, which depends on the distribution $\mu$ of the data point $Z_t$. However, it seems that this information maybe unknown in practice, which has also been partially discussed in page 3 of this paper.
3) It is a bit confusing why the same (even better) rate can be achieved in the non i.i.d. setting, because it seems that this setting should be more challenging than the i.i.d setting.
4) For online learning with delayed feedback, the authors follow the framework in Weinberger and Ordentlich (2002), which actually is known to be inefficient in the sense that $d$ instances of a non-delayed online algorithm are maintained. According to the most recent studies on online learning with delayed feedback, I suggest that the authors may develop a specific variant of EWA and FTRL for handling delayed feedback. For example, one possible attempt is to replace $\sum_{s=1}^t c_s$ in FTRL with $\sum_{s=1}^{t-d} c_s$. Moreover, a more detailed literature review on delayed online learning maybe helpful.

**Paper Award:**

No

---

> ### Author Response · Authors · 2024-11-23
>
> Thank you for your careful reading of our paper and positive evaluation! We respond to your questions below.
>
> **Q1: “It seems that the main results can be simply proved by combining…”**
>
> Note that proving this result required the idea of introducing delays into the online algorithm, and thus it cannot be recovered immediately from the results of Theorem 1 of Lugosi & Neu without this modification. It is not trivial that such a simple adjustment would work, see, e.g. the concurrent work of Chatterjee and Sethi that did not use delays and ended up having to make very strong assumptions on the online algorithm and the loss function to obtain their results.
>
> **Q2: “The online learner needs to know $\mu$”**
>
> This is indeed the case, and is one of the more subtle properties of the reduction framework of Lugosi & Neu. As discussed extensively in their work (and our page 3), the generalization game is a purely abstract analytical tool: it is never actually played in practice, and is only used to decompose the generalization error into the sum of the regret of some online learner and a martingale. Being an abstract construction that is only used within the proofs, the online learner is allowed to know everything about the problem setting, including the distribution $\mu$ of data points, the mixing time, etc. Please refer to Lugosi & Neu for a more detailed discussion of this.
>
> **Q3: “It is unclear why the same or better rates can be achieved as in the i.i.d. case”**
>
> We are not sure what this comment refers to: as expected, our rates are all worse than what can be achieved in the i.i.d. case. For instance, Corollary 12 features an additional factor of $\tau$ on top of the classic PAC-Bayesian bound that can be recovered from the conversion scheme of Lugosi & Neu. The rate of Corollary 11 is worse than the $\sqrt{1/n}$ rate typically achieved in i.i.d. settings.
>
> **Q4: “The reduction of Weinberger and Ordentlich is inefficient”**
>
> You are entirely correct: there are more efficient methods known for online learning with delays. However, notice again that the beauty of the online-to-PAC machinery is that the online learning algorithm never needs to be implemented in reality, as it is only used within the proof (as per our response to your Q2 above). Thus, whether or not the online method is computationally efficient or not is immaterial. On the other hand, the regret guarantees achieved by the reduction of Weinberger & Ordentlich are already minimax optimal, and thus one cannot expect to improve them without making further assumptions about the losses (which we avoided in this paper). We will explain this in more detail in the final version, and provide a more complete discussion on the delayed online learning literature.

---

> > ### Comment · Reviewer_ocxV · 2024-11-25
> >
> > Thank the authors for your detailed responses. My concerns have almost been addressed. However, I am a bit confused about whether the generalization error bounds in this paper can be achieved by a practical algorithm if the generalization game is utilized as a purely abstract analytical tool. Or, is such a practical algorithm not necessary for the generalization analysis of statistical learning?

---

> > > ### Author Response · Authors · 2024-11-27
> > >
> > > Thank you for following up! Your comment that "such a practical algorithm not necessary for the generalization analysis of statistical learning" is indeed correct. Let us clarify below.
> > >
> > > The goal in our work (as in Lugosi & Neu) is to give a bound on the generalization error of a *given statistical learning algorithm* $\mathcal{A}$, and the online learning algorithm is only used *within the analysis* to show that $\mathcal{A}$ generalizes well. This is shown in Theorem 1: the generalization error of $\mathcal{A}$ equals the regret of the online learner against $\mathcal{A}$ (up to a perturbation term which is close to a martingale in our construction). As such, in order to show that $\mathcal{A}$ generalizes well, we only need to demonstrate that *there exists* an online learning algorithm with guaranteed regret against $\mathcal{A}$. This online algorithm does not need to be practical since it is never executed in the "real world". In contrast, $\mathcal{A}$ can be any algorithm we wish to analyze, including the most practical ones that one can think of.
> > >
> > > We hope that this fully answers all of your questions, but do let us know if you need us to clarify further.

---

> > > > ### Comment · Reviewer_ocxV · 2024-11-28
> > > >
> > > > Thanks for the authors' clarification. I have no concern and increase my score.

---

### Official Review · Reviewer_cL1W · 2024-11-11
**Elegant extension of Lugosi and Neu (2023) to non-iid processes (albeit mostly combination of existing techniques)**

**Rating:** 7
**Confidence:** 4

**Review:**

Summary of contributions: This paper extends the framework of deriving (Bayesian-style) generalization bounds through a reduction to online learning that was first introduced by Lugosi and Neu (2023) to non-iid processes that have sufficiently weak temporal dependency with respect to how the generalization gap is evaluated (see Assumption 1; essentially a weaker version of $\beta$-mixing). To handle temporal dependency, the reduction is now carried out to online learning with $d$-delayed feedback (Proposition 5 and Theorem 6). Corollaries for geometric and algebraic mixing processes and extensions of the FTRL family with incorporated $d$-delay are then presented in Section 4. As outlined in Section 4, there is a natural and expected tradeoff in the choice of $d$: the higher the value of $d$, the lower the error arising from dependency (Assumption 1 and Proposition 5) but the higher the regret bound of the delayed online learning algorithm. In the main application of the bounds to geometric and algebraic mixing processes (Section 4.2), the mixing parameter dictates the choice of $d$. Finally, in Section 5 an extension of the framework to dynamic hypotheses that incorporate memory is presented, building on the works of Erengis et al (2022 and 2024).

Strengths of submission:
- I found the original work of Lugosi and Neu (2023) to be very novel and innovative and consider this to be a nice extension of their framework to non-iid data. Assumption 1 (indeed, even the slightly stronger $\beta$-mixing assumption) is very natural.
- Although the results are largely a careful combination of existing techniques (Assumption 1 directly implies the generalization bound in expectation, i.e. Proposition 5; Lemma 7 uses the classical blocking technique of Yu et al (1994), and the applications to explicit delayed online learning algorithms and mixing processes take the form of corollaries), the framework is simple, elegant and general which I consider to be a positive.
- The paper is overall well-written, well-structured and well-contextualized with related work. As mentioned in Section 6 it is considerably simpler and more general than some related recent approaches to generalization bounds under non-iid data.

Weaknesses of submission:
- I think the statement that Assumption 1 differs/is weaker than the usual $\beta$-mixing assumption is somewhat overblown given that the authors ultimately only present applications of their results to $\beta$-mixing processes in Section 4.2. I could not think of any natural example of a process that does not satisfy $\beta$-mixing but would satisfy Assumption 1. It would be great if the authors could provide such an example, if it exists, or try to think of one, and it would make me even more enthusiastic about this submission if so.
- As mentioned above, given that the results mostly combine existing techniques, I would say that the technical novelty is somewhat limited (I consider this to be a minor weakness, though).
- While the paper is well written overall, there were a couple of aspects where I thought the presentation could be enhanced. First, while I appreciate the way in which Sections 3.1 and 3.2 are structured such that the reader can make direct comparisons between them, I think Theorem 2 could be expanded to include both the in-expectation and high-probability bound, so that a reader who may not have been familiar with Lugosi and Neu (2023) already could easily compare the in-expectation iid bound with Proposition 5 and Theorem 6 respectively. (I understand this easily follows from text that is already in the paper, but still think this could be presented slightly more cleanly). Second, I think Section 4.2 could benefit from mentioning implications to common examples such as Markov chains and hidden Markov models -- both of these fall under $\beta$-mixing processes but I think it would be helpful for the reader to see them explicitly written out.

Questions for authors:
- This is just out of curiosity, but you mention in Section 6 the concurrent work of Chatterjee and Sethi (2024) that reduces to standard online learning algorithms (without delay) but does need to make strong assumptions about the loss function/online learning algorithm. I was curious about whether you believe there is any scope to make a similar reduction instead making stronger assumptions on the process. I was also curious about whether there are non-mixing, e.g. periodic processes that could be handled given that there do exist results in the literature concerning learning, or concentration without mixing in these cases (albeit for simpler settings/special cases; see https://projecteuclid.org/journals/annals-of-applied-probability/volume-14/issue-2/Optimal-Hoeffding-bounds-for-discrete-reversible-Markov-chains/10.1214/105051604000000170.full and https://proceedings.mlr.press/v75/simchowitz18a.html).
- Relatedly, one other shortcoming of Corollary 10 and 11 (and its further applications to Corollaries 12 and 13) is that it does require knowledge of the mixing parameters $\tau$ or $r$. I wondered if there is any scope to design online learning algorithms that adapt the delay parameter to data in a "model selection" style. This seems like it could be possible using online model selection routines as a black box.

***UPDATE AFTER REBUTTAL***
I am happy with the authors' response and I have slightly increased my score. Please see my follow-up comment for details on suggestions to include in camera-ready version.

**Paper Award:**

No

---

> ### Author Response · Authors · 2024-11-23
>
> Thank you for your thoughtful review and many interesting questions! We respond to them below.
>
> **Weakness 1: The relation between $\beta$-mixing and our assumption.**
>
> This is a great question! While on the first look, our assumption can look like a small “cosmetic” improvement over $\beta$-mixing, it can be often much weaker under conditions that are not very artificial. For example, consider the following simple case. Let the loss function be defined as $\ell(w,Z_t) = \ell(w,Z_t’) + \alpha\dot\varepsilon_t$ where $Z_t’$ is an i.i.d. sequence, $\varepsilon_t$ is sampled from a bounded $\beta$-mixing process and $\alpha$ is a small constant. Now, for any $w$, $\ell(w,Z_t)$ is clearly a $\beta$-mixing process inheriting the properties of $\varepsilon_t$, independently of the choice of $\alpha$. In contrast, the mixing rate as defined in our Assumption 1 improves linearly with $\alpha$, and vanishes as this parameter approaches zero. In other words, $\beta$-mixing conditions are very strict in that they require mixing in terms of the entire distribution of the loss, and ignores the “scale” at which nonstationarity impacts the outcomes. (I.e., nonstationarity in the lower-order bits is treated the same as nonstationarity in the higher-order bits.) We hope that this simple example illustrates clearly the differences between the two assumptions. We will include it in the final version of the paper. To be clear, we are not claiming that our assumption would be a major improvement over $\beta$-mixing, but we nevertheless believe that it is a much more natural and straightforward assumption that suffers from less pathologies.
>
> **Weakness 2: limited technical novelty.**
>
> Thank you for affirming that this is a “minor weakness”. We believe that the simplicity of our approach is the main strength of this work, and that the novelty lies in identifying a problem formulation that allows the use of simple techniques to address an otherwise complex problem that has been studied quite heavily before.
>
> **Weakness 3: recommendations on presentation.**
>
> Thank you for these suggestions, we will take them into serious consideration when preparing the final version!
>
> **Q1: beyond mixing processes**
>
> These are great suggestions! We find it plausible that our techniques can be extended to the settings studied in these two papers, but perhaps using delays might not be the perfect approach in these cases. For instance, when dealing with periodic processes, it feels more natural to work with online algorithms that change their policies every $d$ rounds, with $d$ set according to the periodicity of the process (to make sure that the total loss of the online learner behaves like a martingale). We will mention these as potential extensions in the final version.
>
> **Q2: knowledge of mixing parameters**
>
> Note that for the bound to be valid, one does not need to run the online learning algorithm, and in fact the online algorithm is allowed to know the mixing parameters. (This is the beauty of the online-to-PAC framework: the online algorithm is an abstraction that is only used within the analysis, and is never executed in practice.) Thus, evaluating the bounds in practice boils down to estimating the mixing parameters, which is a difficult problem with a large literature in its own right. (Similarly, the question you ask is a very interesting one, but ultimately not relevant to our setting.)

---

> > ### Comment · Reviewer_cL1W · 2024-11-23
> > **Thanks for the detailed responses**
> >
> > Dear authors,
> >
> > Thank you very much for your detailed explanations. The example contextualizing how your assumption is a bit weaker than $\beta$-mixing is helpful. I was aware that the online algorithm is used as an analysis tool rather than an explicit algorithm, but had forgotten that nuance in the context of knowing the mixing parameter. (As you say, the ability to evaluate bounds in practice for mixing processes is an independently non-trivial question but somewhat out of scope for this paper.)
> >
> > I will increase my score slightly based on your response. Please try to include the content of your response in some form into the camera-ready version.

---

### Official Review · Reviewer_rtYW · 2024-11-16

**Rating:** 7
**Confidence:** 4

**Review:**

This paper derives generalization bounds for non-iid data that are sampled from a stationary mixing process. More specifically, the assumption on the samples Z_1,…,Z_t,… is that, for any time step t, the expectation of the gap between the true loss of any hypothesis w and the loss of w on the sample Z_t, conditioned on the past d samples, is at most phi_d (which is a function that generally decreases with d, with common choices having exponential or polynomial decay). The more common \beta-mixing assumption (which does not involve the loss function) implies this condition when the losses are bounded.

The technique is an extension of the online-to-PAC conversion of Lugosi and Neu (2023): this work introduced an online learning game where in each step t, the online learner picks a distribution over hypotheses, the adversary picks the cost function of this step to be determined by the datapoint Z_t, the online learner incurs the average cost of their strategy and sees the datapoint Z_t.  Lugosi and Neu showed that the generalization error of a learning algorithm A can be decomposed into the average regret of an online learner as compared to the fixed strategy of playing the distribution produced by A in each step, and the total cost incurred by the online learner. Choosing different online learners can then give us different generalization bounds.

In this work, the authors show that changing the online learning game to introduce a delay d (i.e. the learner sees the Z_{t-d+1} datapoint and not Z_t) allows one to derive a similar decomposition of the generalization gap where the first term is the average regret of a d-delayed online learner and the second can be bounded by phi_d (notice that this implies that the generalization gap is bounded by these terms for all d simultaneously).

Choosing d to be roughly log(n) for exponentially decaying phi_d and instantiating the bound with standard online learners such as MW and FTLR gives us generalization bounds that hold for non-iid data, and match the known bounds for iid data with an extra multiplicative sqrt{log(n)} factor. Similar results are retrieved for polynomially decaying phi_d.

The authors additionally consider an extension to loss functions that depend on the whole sequence of datapoints so far, which captures dynamical predictors, and show that if the losses satisfy some bounded-memory conditions, this case can be reduced to the previous one.

Overall, the techniques do not depart significantly from Lugosi and Neu 2023, but I think that it is surprising that this general framework can handle non-iid data, for which most generalization bounds seem to follow tailored approaches. I think that this paper demonstrates the versatility of the online-to-PAC generalization approach and it would be interesting to the ALT community.


Questions:
- To upper-bound the d-delayed online learning regret, the authors use a general transformation of the regret of a (non-delayed) standard online learner, e.g. MW. Are there online learners that are optimized for the delayed setting and can give us better bounds, perhaps shaving some log factors? Or is there a lower bound that implies that this is the best bound we can achieve anyway?
- Is there an inherent difficulty in achieving fast rates when the training error is zero, for the non-iid case? Or is it possible that one can use better online learners and similar tricks as in the online-to-PAC conversion for iid data that allows us to get fast rate convergence?

Typo:
Lemma 8: “is a concave in y [..]”

**Paper Award:**

No

---

> ### Author Response · Authors · 2024-11-23
>
> Thank you for your positive review, and for appreciating the simplicity and generality of our approach! We respond to your questions below.
>
> **Q1: "are there better algorithms for delayed online learning?"**
>
> Following Weinberger & Ordentlich (2002), Joulani et al (2013) have shown that the result of Lemma 8 is minimax optimal, and in particular the best that one can hope for without further assumptions on the loss is a regret of $O(\sqrt{dT})$. One can get better regret bounds under more specific assumptions about the losses, for instance with the i.i.d. assumption (see the nice Table 1 of https://proceedings.mlr.press/v28/joulani13.pdf). Notice however that this is precisely the type of assumption that we wanted to avoid in our work. We will mention this explicitly in the final version.
>
> **Q2: "can you get fast rates?"**
>
> The fast rate results of Lugosi & Neu (Section 3.1.3) are non-trivial applications of the online-to-PAC framework. The main difficulty to extend their results to our setting is that the regret bound they take as starting point is a data-dependent one, and is not simply a function of the horizon $n$. Therefore the black-box machinery of Lemma 8 cannot be applied directly to get a delayed regret bound of $dR(n/d)$. The probabilistic term in the right-hand side however could inherit of the nice property given by Lemma 28 of Neu and Lugosi at a price of the additive factor phi_d which is of O (log n / n) in the case geometric mixing, therefore yielding a fast rate. We find it possible that the data-dependent bound of Lugosi & Neu can be adapted to our setting at the cost of a more complex analysis, but in the present submission we preferred to focus on simplicity.

---

### Author Rebuttal · Authors · 2024-11-23

We thank the reviewers for their time and constructive feedback on our submission, which we will incorporate to improve our paper. We respond to each reviewer separately below.

---

### Meta-Review · Area_Chair_dr3u · 2024-12-13

**Recommendation:** Accept
**Confidence:** 4

**Metareview:**

This paper establishes generalization bounds for non-i.i.d. data sampled from a stationary mixing process, leveraging the online-to-PAC conversion framework introduced by Lugosi and Neu (2023). The reviewers all appreciate the contributions of this work.

To improve the discussion, it would be good for the authors to elaborate on the relationship between their results and prior work by Kuznetsov et al. in particular,  the following:
Discrepancy-Based Theory and Algorithms for Forecasting Non-Stationary Time Series. Ann. Math. Artif. Intell., 88(4): 367–399 (2020).
Time Series Prediction and Online Learning. COLT 2016: 1190–1213.
Generalization Bounds for Non-Stationary Mixing Processes. Mach. Learn., 106(1): 93–117 (2017).
Generalization Bounds for Time Series Prediction with Non-Stationary Processes. ALT 2014: 260–274.

This discussion would help clarify how the proposed methods relate to existing approaches.

**Paper Award:**

No